# Provocation and Localization of Arrhythmogenic Triggers from Persistent Left Superior Vena Cava in Patients with Atrial Fibrillation

**DOI:** 10.3390/jcm12051783

**Published:** 2023-02-23

**Authors:** Kentaro Minami, Kohki Nakamura, Eiko Maeno, Keitaro Iida, Ikuta Saito, Taiki Masuyama, Yoshiyuki Kitagawa, Toshiaki Nakajima, Yosuke Nakatani, Shigeto Naito, Shigeru Toyoda, Milan Chovanec, Jan Petrů, Jan Škoda, Koji Kumagai, Petr Neužil

**Affiliations:** 1Department of Cardiovascular Medicine, Dokkyo Medical University, Mibu 321-0293, Tochigi, Japan; 2Gunma Prefectural Cardiovascular Center, Maebashi 371-0004, Gunma, Japan; 3Department of Cardiology, Na Homolce Hospital, 15030 Prague, Czech Republic; 4Department of Cardiovascular Medicine, Tohoku Medical and Pharmaceutical University, Sendai 983-8536, Miyagi, Japan

**Keywords:** persistent left superior vena cava, atrial fibrillation, catheter ablation

## Abstract

Background: Although pulmonary vein isolation (PVI) is an established procedure for atrial fibrillation (AF), non-PV foci play a crucial role in AF recurrence. Persistent left superior vena cava (PLSVC) has been reported as critical non-PV foci. However, the effectiveness of provocation of AF triggers from PLSVC remains unclear. This study was designed to validate the usefulness of provoking AF triggers from PLSVC. Methods: This multicenter retrospective study included 37 patients with AF and PLSVC. To provoke triggers, AF was cardioverted, and re-initiation of AF was monitored under high-dose isoproterenol infusion. The patients were divided into two groups: those whose PLSVC had arrhythmogenic triggers initiating AF (Group A) and those whose PLSVC did not have triggers (Group B). Group A underwent isolation of PLSVC after PVI. Group B received PVI only. Results: Group A had 14 patients, whereas Group B had 23 patients. After a 3-year follow-up, no difference in the success rate for maintaining sinus rhythm was observed between the two groups. Group A was significantly younger and had lower CHADS2-VASc scores than Group B. Conclusions: The provocation of arrhythmogenic triggers from PLSVC was effective for the ablation strategy. PLSVC electrical isolation would not be necessary if arrhythmogenic triggers are not provoked.

## 1. Introduction

Because several studies have demonstrated that pulmonary veins (PVs) are the primary site of ectopic beats initiating atrial fibrillation (AF), PV isolation (PVI) has been established as a cornerstone ablation procedure for curing AF [1,2]. In contrast, non-PV ectopic beats play a crucial role in AF recurrence after PVI [3,4]. Non-PV ectopic beats initiating AF have been reported, such as the superior vena cava (SVC) [5], left atrial (LA) posterior wall [6], coronary sinus (CS) [7], ligament of Marshall (LOM) [8]. The persistent left superior vena cava (PLSVC) is the embryological precursor of the LOM, which has also been reported as non-PV foci, with a reported incidence of 0.5% in the general population [9,10]. Although several studies have reported the positive effects of electrical isolation of the PLSVC [11,12], whether electrical isolation is necessary for all PLSVC cases remains unclear. Typically, non-PV foci triggers can be elicited using standard induction protocols, including cardioversion of spontaneous or induced AF and infusion of high-dose isoproterenol [13]. However, few reports have scrutinized the arrhythmogenicity of the PLSVC using induction protocols, and the effectiveness of provoking triggers from PLSVC remains unclear. This study was designed to detail this multicenter experience in investigating AF triggers from the PLSVC and the effectiveness of electrical isolation of PLSVC.

## 2. Methods

### 2.1. Patients Population

This multicenter retrospective study included 37 patients with PLSVC from three centers who underwent catheter ablation for symptomatic drug-refractory AF from January 2005 to December 2019. The study protocol was approved by the Institution Research and Ethics Committee of all involved institutions and conducted according to the Declaration of Helsinki. Demographic data, including patient, background, procedural details, computed tomography (CT) imaging studies, follow-up visits, and echocardiograms were obtained from the medical records. All patients underwent preprocedural imaging using 64-slice contrast CT scanning (SOMATOM Definition, SIEMENS) to evaluate the presence of the PLSVC and coexisting heart disease. 

### 2.2. Pulmonary Vein Isolation

All anti-arrhythmic drugs (AADs) were discontinued for at least five half-lives before the electrophysiological test and ablation procedure except for amiodarone, which was discontinued at least 7 days before ablation. A transesophageal echocardiogram was performed to exclude left atrial thrombus and investigate the other congenital heart disease. The procedure of PVI has been previously described in detail [14]. PVI was performed using an ipsilateral double-circular mapping technique guided by fluoroscopy and angiography in all patients undergoing the first ablation. Successful electric PVI was confirmed by the entrance and exit block as registered by the circular mapping catheter (Lasso, Biosense Webster Inc., Irvine, CA, USA) in the respective pulmonary veins. In patients undergoing redo ablation, PVI was assessed by pacing maneuvers using a circular mapping catheter, and PV isolation was performed if PV was electrically reconnected. 

### 2.3. Provocation and Localization of Arrhythmogenic Triggers from Persistent Left Vena Cava

Following the PVI procedure, the electrophysiological study was performed in all patients to provoke the triggered activity from the PLSVC (Figure 1). Venography of the PLSVC was performed using contrast injection from a long sheath (SL0 8Fr, St. Jude Medical) in the PLSVC, and the circumferential mapping catheter was introduced retrogradely through the PLSVC using intracardiac echocardiography (ICE) guidance (Figure 2 and Figure 3). Double potentials that consist of left atrial far-field potentials and sharp local PLSVC near-field potentials were confirmed. To evaluate AF triggers from the PLSVC, persistent or pacing-induced AF was cardioverted, and re-initiation of AF was monitored under high-dose isoproterenol infusion (up to 20 µg/min). If two or more spontaneous initiations of AF by a premature complex were observed, the PLSVC was regarded as arrhythmogenic as it has an AF trigger (Figure 4). According to the electrophysiological study, patients were divided into two groups: those whose PLSVC had AF triggers (Group A; Arrhythmogenic PLSVC) and those whose PLSVC did not have AF triggers (Group B; PLSVC without triggers). We also investigated the location of the triggers initiating AF in the PLSVC. During the mapping of the triggers from the PLSVC, a circular mapping catheter and a duodecapolar catheter were placed in the PLSVC from the dilated CS (Figure 3). The mid-PLSVC was defined as the point between the left superior and inferior pulmonary veins. The portion close to the CS from the mid-PLSVC was defined as the proximal PLSVC, and the beyond the mid-PLSVC was defined as the distal PLSVC.

### 2.4. Catheter Ablation for Persistent Left Vena Cava

The electrical isolation of the PLSVC was performed if the PLSVC had AF triggers to prevent the arrhythmogenic triggered activity from conducting to the atria. Before completing the electrical isolation of the PLSVC, premature ectopy with the earliest activation in the PLSVC was evaluated during the mapping procedure. If the location of the AF trigger emerged from the distal-PLSVC, electrical isolation of the PLSVC was achieved through a circumferential ablation at the level of the mid-PLSVC (between the left superior pulmonary vein and inferior pulmonary veins). If the location of the AF trigger was in the proximal-PLSVC, focal ablation or segmental isolation was performed targeting the trigger site of the PLSVC and LA connection. The power used for ablation in the PLSVC was 15–20 W and was restricted to a maximum time of 20 s for each ablation point with a flow rate of 17 mL/min, targeting a maximum temperature of 43 degrees. Successful isolation of PLSVC was defined as the achievement of either exit and entrance blocks or electrical disconnection between the PLSVC and LA, which was confirmed by the reversal of the activation sequence in the CS during LA appendage pacing. If disconnecting the electrical connection between the PLSVC and LA was difficult, applications from the LA were added as needed. Maximal output pacing through the ablation catheter was delivered to confirm left phrenic nerve was captured, but the point was not ablated. After the ablation to the PLSVC, a high-dose isoproterenol infusion was performed to verify that all triggers were isolated or eliminated. 

### 2.5. Follow-Up

All patients visited the clinic and underwent 12-lead electrocardiogram recording 1, 3, 6, 12, 15, 18, 21, 24, 30, and 36 months after the procedure. All patients were equipped with an event recorder or 24 h electrocardiogram to analyze symptoms suggestive of arrhythmia recurrence. Antiarrhythmic drugs were continued for 3 months during the blanking period. Any documented episodes of atrial tachyarrhythmia lasting >30 s after 3 months of the blanking period were considered a recurrent arrhythmic event (late recurrence)—the occurrence of any serious adverse events within the first 30 days after the procedure was recorded. A standard definition of serious adverse events was used: any ablation- or procedure-related untoward medical occurrences leading to death or a deterioration in the patient’s health that resulted in life-threatening illness or injury, permanent impairment of a body structure or a body function, and/or prolonged inpatient hospitalization or medical or surgical intervention to prevent life-threatening illness or injury or permanent impairment to a body structure or a body function. 

### 2.6. Data Analysis

The data are presented as means ± standard deviations. The data were compared using Student’s *t*-test, the chi-square test, Fisher’s exact test, and the Kalan-Meier curve analysis with a long-rank test as appropriate, using Statistical Package for the Social Sciences, version 24.0 (IBM Corp., Armonk, NY, USA). *p*-value equal to or <0.05 was used to denote statistical significance.

## 3. Results

### 3.1. Patient Characteristics

In this study, 37 patients who had PLSVC with symptomatic drug-refractory AF who underwent catheter ablation were included. Patient characteristics are shown in Table 1. The mean age of the patients was 61 ± 11 years; 70% (26/37) were males, and the mean duration of AF was 31 ± 24 months. Paroxysmal AF was observed in 81% (30/37) of the patients, and 19 % (7/37) had persistent AF. The mean LA size was 37 ± 5.5 mm, and the left ventricular ejection fraction was 63 ± 8.0%. The mean CHA2DS2-VASc score was 2.1 ± 1.7.

### 3.2. Provocation of Arrhythmogenic Triggers from Persistent Left Vena Cava 

Electrical PVI was complete for all patients. Following the PVI procedure, all 37 patients underwent the provocation of arrhythmogenic triggers from the PLSVC. Through electrophysiological examination, 14 patients were determined to have AF triggers from the PLSVC (Group A), whereas 23 had no remarkable AF triggers from the PLSVC (Group B). The comparison of the two groups showed no difference in the duration of AF, left ventricular ejection fraction, and the diameter of the PLSVC. However, the patients in Group A were significantly younger (53 ± 8.8 tears vs. 72 ± 5.9 years; *p* < 0.001) and had significantly lower CHA2DS2-VASc scores (0.77 ± 1.01 vs. 3.11 ± 1.49; *p* < 0.001) (Table 2). The procedure time (minutes) was significantly longer in Group A. (186 ± 58 min vs. 137 ± 38 min; *p* < 0.001). The occurrence of any complications did not differ between the two groups.

### 3.3. Localization of Triggers and Electrical Isolation of Persistent Left Superior Vena Cava

All 14 patients with AF triggers from the PLSVC underwent ablation procedures to the PLSVC. The triggers appeared from the distal PLSVC in 11 patients (81.3%) and from the proximal PLSVC in three patients (18.7%). The 11 patients who had AF triggers from the distal-PLSVC underwent electrical isolation at the level of the mid-PLSVC by circumferential applications. The three patients with AF trigger from the proximal-PLSVC underwent focal ablation or segmental isolation to the trigger site. No significant difference in the success rate of PLSVC isolation was observed; however, the procedure time was significantly longer in patients with AF trigger emerging from the proximal-PLSVC. All three patients with AF trigger from the proximal-PLSVC required additional ablation to the LA to disconnect the electrical connection between the PLSVC and LA (Table 3). No complications associated with the electrical isolation of the PLSVC were observed.

### 3.4. Clinical Recurrence

After 1210 ± 110 days of follow-up, the 3-year Kaplan-Meier estimate for freedom from atrial arrhythmias was not significantly different between Group A and Group B (71.4% vs. 82.6%; *p* = 0.607) (Figure 5).

## 4. Discussion

The primary findings of this study are as follows: (a) the provocation and localization of the triggered activity initiating AF from the PLSVC were important to determine the ablation strategy; (b) the electrical isolation of the PLSVC, which had AF triggers associated with long-term freedom from AF at the 3-year follow up; (c) trigger-free PLSVC may not require electrical isolation; (d) the electrical isolation of the PLSVC was performed safely without serious complications; and (d) the location of the arrhythmogenic triggers in the PLSVC was important. If the arrhythmogenic triggers emerge from the proximal PLSVC, the electrical isolation of the PLSVC was not simple. Further ablations might be required from the LA because of the disconnection of various electrical connections between the PLSVC and LA. 

The role of the PLSVC in initiating AF was first identified by Hsu et al. [11], and they circumferentially isolated the PLSVC in five patients. Since then, several case series have been published demonstrating the PLSVC as an arrhythmogenic source in AF. Thus, several reports have reported the effectiveness of isolation of PLSVC [15]; however, the usefulness of provocation and localization of the arrhythmogenic triggers from the PLSVC remains unclear. This study suggests the usefulness of the provocation and localization of the arrhythmogenic triggers from the PLSVC. Furthermore, this is the first paper to show that the isolation of the PLSVC without AF triggers is not necessary. Distinguishing between triggered and non-triggered PLSVCs may help in selecting ablation treatment strategies.

A prior study has reported that the electrical isolation of the PLSVC was associated with increased complications, such as cardiac tamponade [16]. Therefore, identifying cases in which PLSVC isolation is unnecessary would be clinically valuable if the arrhythmogenic triggers initiating AF from the PLSVC can be provoked. Furthermore, in this study, no significant difference in the 3-year outcome of AF-free survival was observed between the two groups. The provocation of AF triggers from the PLSVC may be able to separate cases that require the electrical isolation of the PLSVC from those that do not. The right SVC contains embryonic sinus venous tissue capable of spontaneous firing [17], and also has been shown as one of the most important sites of origin for non-PV triggers [5,18]. The frequency of AF trigger appearance from SVC is reported to be about 2% [19]. In comparison, the frequency of AF Trigger appearance from the PLSVC was higher than that from the right SVC.

There are several reasons for the arrhythmogenicity of the PLSVC [19]. In the embryonic heart, bilateral pace-making areas are present near the sinus horns and common cardinal veins [20]. While the right side takes over the cardiac pace-making function as the sinoatrial node, the persistence of the left common cardinal vein as the LSVC may be associated with the continuing presence of pacemaker tissue and hence ectopic pacemaker activity. Furthermore, overlapping muscle sleeves into the PLSVC has been demonstrated to be a source of abnormal triggers and electrical potentials [21]. The PLSVC is the embryonic remnant of the LOM. The LOM comprises a vestigial fold in the back of the left auricle, extending from the CS to the orifice of the left superior pulmonary vein (LSPV), and contains the vein of Marshall and muscle sleeves from the CS [22]. Repetitive activities from the LOM initiating AF have been reported [23]. In the current study, the group of patients with PLSVC having AF triggers were significantly younger and had lower CHADS2-VASc scores. On the contrary, the patients with PLSVC without triggers may have the usual age range and background disease of patients of AF. If a young AF patient without underlying disease has the PLSVC, electrical isolation of the PLSVC may be necessary because the PLSVC might have arrhythmogenic triggers.

In this study, the electrical isolation of the PLSVC was performed safely, and no increased complications associated with the procedure were observed. If the provoked arrhythmogenic triggers emerged from the distal-PLSVC, the electrical isolation procedure of the PLSVC was relatively simple. Circumferential ablation to the PLSVC at the level between the left superior PV and left inferior PV was performed the same as right SVC electrical isolation. However, if the triggers emerged from the proximal-PLSVC, the isolation of the PLSVC was not straightforward because the paroxysmal-PLSVC has various electrical connections between the LA and PLSVC [21]. Furthermore, the ablation in the proximal CS can be associated with a higher risk of atrioventricular (AV) block. With the PLSVC, the relationship between the compact AV node and CS can be distorted [24]. In our study, three patients needed additional applications to the lateral LA to disconnect the electrical connection between the LA and PLSVC.

## 5. Limitations

This study has several limitations. First, it was a non-randomized observational retrospective study with a small sample size. Second, the results were pooled from three centers and multiple operators with differences in the extent of ablation performed. However, the protocol for the provocation of arrhythmogenic triggers of non-PV foci was similar, and no significant difference in the 3-year outcomes of AF-free survival was observed between the two groups. Finally, we cannot rule out asymptomatic subclinical AF recurrence in patients as none of them had a cardiac implantable electronic device. Furthermore, we cannot rule out the possibility of having a more significant arrhythmia recurrence on longer follow-ups. The role of supplementally ablation procedures remains controversial and needs further investigation in ablation technology and patient selection.

## 6. Conclusions

In patients with AF and PLSVC, the provocation and localization of the ectopic triggers from the PLSVC were effective for catheter ablation. PLSVC electrical isolation would not be necessary if the arrhythmogenic triggers were not provoked.

## Figures and Tables

**Figure 1 jcm-12-01783-f001:**
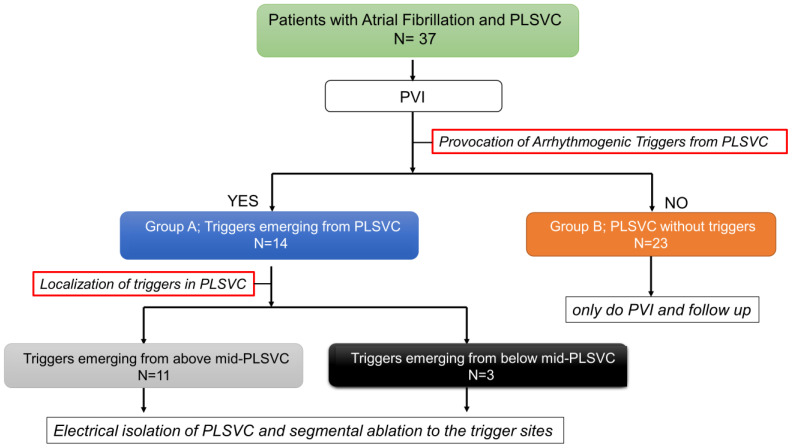
Diagram Flow Showing Patient Treatments and Procedures. PLSVC, persistent left vena cava; PVI, pulmonary vein isolation.

**Figure 2 jcm-12-01783-f002:**
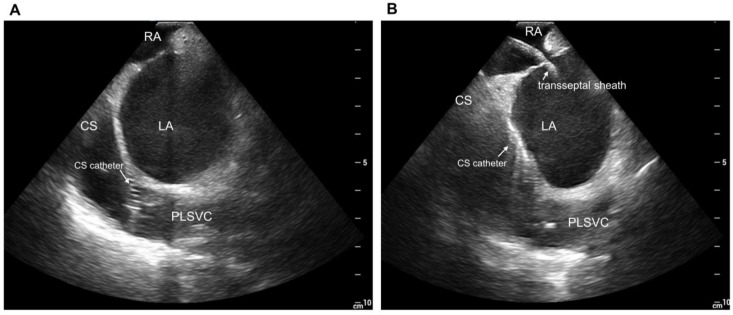
Intracardiac echocardiographic views. (**A**): Duodecapolar catheter inside the PLSVC from dilated CS. (**B**): View of the PLSVC after the transseptal puncture. CS, coronary sinus; LA, left atrium; RA, right atrium; PLSVC, persistent left superior vena cava.

**Figure 3 jcm-12-01783-f003:**
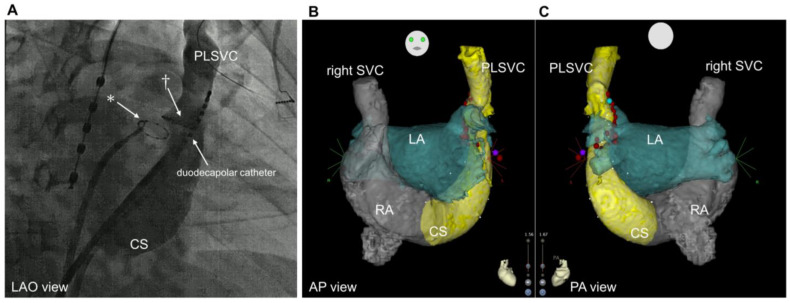
Fluoroscopic images and computed tomography. (**A**): Coronary sinus venography performed with a long sheath placed within the dilated coronary sinus (LAO view). * Circular mapping catheter placed in the left inferior pulmonary vein † Circular mapping catheter placed in the PLSVC. (**B**): Preprocedural CT imaging. anteroposterior view. (**C**): Posteroanterior view. CS, coronary sinus; PLSVC, persistent left vena cava; LAO, left anterior oblique; SVC, superior vena cava; LA, left atrium; RA, right atrium; CT, computed tomography.

**Figure 4 jcm-12-01783-f004:**
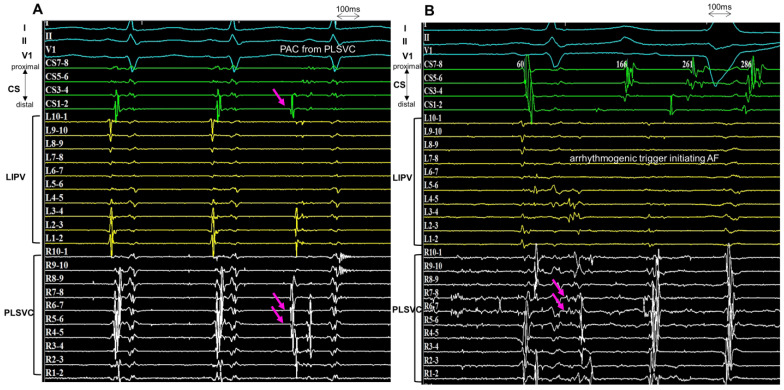
Arrhythmogenic trigger from PLSVC. (**A**): The circular mapping catheters are positioned within the LIPV and PLSVC. During the first two sinus beats, the two components (left atrial far-field and PLSVC potential) are fused. During the ectopic rhythm (the third beat), the sharp potential of the PLSVC precedes the surface ECG by 60 ms. The atrial activation sequence is from distal to proximal on the CS during the PAC. (**B**): During AF initiation, the sharp potential of the PLSVC precedes the PAC, and the PAC arising from the PLSVC initiates the AF. CS, coronary sinus; PLSVC, persistent left vena cava; LIPV, left inferior pulmonary vein; PAC, premature atrial contraction; AF, atrial fibrillation.

**Figure 5 jcm-12-01783-f005:**
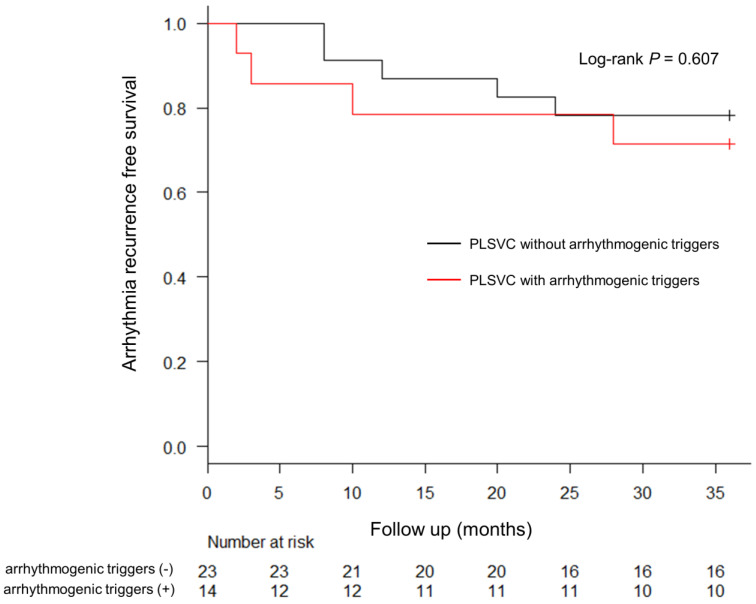
Kaplan-Meier Estimates of 3-Year Freedom from Atrial Arrhythmia Recurrence. The recurrence of atrial arrhythmia in both groups after the last ablation procedure is shown. The electrical isolation of PLSVC was determined based on the electrophysiological study for the provocation of arrhythmogenic triggers initiating AF. No significant difference in arrhythmia recurrence was observed between the two groups. PLSVC, persistent left superior vena cava; AF, atrial fibrillation.

**Table 1 jcm-12-01783-t001:** Patient Characteristics.

Characteristics	*N* = 37
Age (years) (mean ± SD)	78 ± 8.1
Males, *n* (%)	26 (70%)
Type of AF	
paroxysmal	30 (81%)
persistent	7 (19%)
Duration of AF (months)	31 ± 24
Left atrium diameter, (mm)	37 ± 5.5
Left ventricular ejection fraction, (%)	63 ± 8.0
CHA2DS2-VASc score	2.1 ± 1.7
Congestive heart failure	2 (5%)
Hypertension	19 (51%)
Diabetes	8 (21%)
Stroke	2 (5%)
Coronary artery disease	3 (8%)
Antiarrhythmic drug	9 (24%)

AF, atrial fibrillation.

**Table 2 jcm-12-01783-t002:** Provocation of Arrhythmogenic Triggers from PLSVC.

	Group A (*n* = 14)	Group B (*n* = 23)	
	Arrhythmogenic Triggers (+)	Arrhythmogenic Triggers (−)	*p*-Value
Age (years)	53 ± 8.8	72 ± 5.9	<0.001
Male, *n* (%)	10 (71%)	15 (62%)	0.98
LA diameter (mm)	34.3 ± 5.1	40.7 ± 4.8	0.0011
Paroxysmal AF	12 (86%)	18 (78%)	0.89
AF duration (months)	33 ± 22	26 ± 20	0.35
Left ventricular ejection fraction (%)	62.6 ± 9.1	61.5 ± 9.2	0.79
Diameter of the PLSVC (mm)	19.1 ± 8.5	20.5 ± 6.3	0.50
Antiarrhythmic drugs, *n* (%)	2, (14%)	4, (17%)	0.83
Amiodarone, *n* (%)	0, (0%)	0, (0%)	-
CHADS2-VASc	0.77 ± 1.01	3.11 ± 1.49	<0.001
Redo RFCA	4 (28%)	4 (17%)	0.71
TIA/Stroke within 30 days	0 (0%)	0 (0%)	-
Cardiac tamponade	0 (0%)	0 (0%)	-
Procedure time (m)	186 ± 58	137 ± 38	<0.001

LA, left atrium; AF, atrial fibrillation; PLSVC, persistent left superior vena cava; RFCA, radiofrequency catheter ablation; TIA, transient ischemic attack.

**Table 3 jcm-12-01783-t003:** Procedural Characteristics for Electrical isolation of PLSVC.

Procedural Characteristics	Location of Arrhythmogenic Triggers (n = 14)		
	Distal-PLSVC (*n* = 11)	Proximal-PLSVC (*n* = 3)	*p*-Value
Electrical isolation of PLSVC, *n* (%)	10 (91%)	2 (67%)	0.89
Diameter of the PLSVC (mm)	18.9 ± 8.5	17.1 ± 2.6	0.37
PLSVC ablation time (min)	15.4 ± 6.5	31.1 ± 9.6	0.0107
Ablation to LA, *n* (%)	1 (9%)	3 (100%)	0.015
Cardiac tamponade	0 (0%)	0 (0%)	-
TIA/Stroke within 30 days	0 (0%)	0 (0%)	-
Left phrenic nerve paralysis	0 (0%)	0 (0%)	-
Redo RFCA	2 (18%)	2 (67%)	0.35

LA, left atrium; PLSVC, persistent left superior vena cava; RFCA, radiofrequency catheter ablation; TIA, transient ischemic attack.

## Data Availability

Not applicable.

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
