# Peer review of "Provocation and Localization of Arrhythmogenic Triggers from Persistent Left Superior Vena Cava in Patients with Atrial Fibrillation"

_jcm, 2023, doi:10.3390/jcm12051783_

Round 1
Reviewer 1 Report
The paper deals with the ablation of triggers in the persistent left v. cava superior in atrial fibrillation. For this purpose, 37 patients were examined retrospectively. Overall, an exciting question that is also well presented and relevant with long follow-up times. What bothers me is the study design. All apt with pers. cave. examined and everyone was checked to see if they had triggers in the left-sided vena cava. Isolation was only performed if triggers were found. And now the group with triggers and ablation was compared with the group without triggers and thus without ablation. No difference was found here. I find that somewhat questionable in terms of its validity, because it doesn't really say anything about whether the ablation has helped or whether the triggers are not clinically relevant at all. It would have been much better in the two groups to carry out the ablation for some and not for others and then to see whether the ablation brings any benefit. Otherwise, I find it easy to understand from the English and the individual parts of the article are also well done.
Author Response
Thank you so much for pointing out a significant issue. I think your point is correct, which is one of this study's limitations. If we were to compare the effects of ablation to triggers more accurately, it would have been more effective to divide all arrhythmogenic PLSVCs into two groups with and without electrical isolation of the PLSVC. The reasons why the study design could not be assembled as you indicated are as follows;
- There were only 14 of 37 PLSVCs with arrhythmogenic triggers in this study.
- All the PLSVC with AF triggers had been ablated because this was a retrospective study.
If we can conduct a prospective study in the future, I would like to design it, as you pointed out. Thank you so much for your advice.

Reviewer 2 Report
The strengths of this paper are the following:
1) novelty- nonPV triggers for patients with paroxysmal and persistent afib are difficult to identify, and only approximately 0.5-1% of patients have PSVC. 37 patients is impressive.
2) not all patients with PSVC required isolation
3) follow up did not support isolating PSVC even when triggers were found
A comparison with SVC triggers in patients without PSVC would be interesting in the discussion
Author Response
Point 1: . The strengths of this paper are the following:
1) novelty- nonPV triggers for patients with paroxysmal and persistent afib are difficult to identify, and only approximately 0.5-1% of patients have PSVC. 37 patients is impressive.
2) not all patients with PSVC required isolation
3) follow up did not support isolating PSVC even when triggers were found
A comparison with SVC triggers in patients without PSVC would be interesting in the discussion
Response 1: Please provide your response for Point 1. (in red)
Thank you so much for suggesting. According to your advice, we have added a comparison of the right SVC triggers in patients without PLSVC in the Discussion part. We have added the following sentence and reference to Discussion and Reference. The number of references has been changed due to the addition of references.
We appreciate your advice.
Additional Sentences: Page 9, line 233-237
The right SVC contains embryonic sinus venous tissue capable of spontaneous firing[17], and also has been shown as one of the most important sites of origin for non-PV triggers[5, 18]. The frequency of AF trigger appearance from SVC is reported to be about 2%[19]. In comparison, the frequency of AF Trigger appearance from the PLSVC was higher than that from the right SVC.
Page 11, line 337-341
- Yeh HI, Lai YJ, Lee SH, Lee YN, Ko YS, Chen SA, Severs NJ, Tsai CH. Heterogeneity of myocardial sleeve morphology and gap junctions in canine superior vena cava. Circulation 2001; 104:3152-3157.
- Tsai CF, Tai CT, Hsieh MH, Lin WS, Yu WC, Ueng KC, Ding YA, Chang MS, Chen SA. Initiation of atrial fibrillation by ectopic beats originating from the superior vena cava: electrophysiological characteristics and results of radiofrequency ablation. Circulation 2000; 102:67-74.
- Santangeli P, Zado ES, Hutchintn MD, Riley MP, Lin D, Frankel DS, Supple GE, Garcia FC, Dixit S, Callans DJ, Marchinski FE. Prevalence and distribution f focal triggers in persistent and long-standing persistent atrial fibrillation. Heart Rhythm 2016; 13:374-382.

Reviewer 3 Report
This study was designed to investigate the triggers of AF from PLSVC and the effectiveness of electrical isolation of these PLSVC.
Firstly, the number of patients was not very large but correlated with the low incidence of persistent PLSVC in the general population. The collection of all these patients should be appreciated.
Remarks :
A. The authors divided the patients into two groups: PLVSC with triggers (group A) and PLVSC without triggers (group B):
1. The percentage of AF induction in the 2 groups was not described. We only know the percentage of AF initiated by triggers from the PLSVC.
2. The percentage of antiarrhythmic drugs in groups A and B is not known.
3. Anti-arrhythmic drugs were discontinued for at least five half-lives, with the exception of amiodarone. Patients in group B are older than those in group A and potentially received the most amiodarone. It would be interesting to know the percentage of patients on amiodarone in each group. Group B potentially has more patients on amiodarone which could explain the lack of triggers from the PLSVC.
B. All patients in group A underwent electrical isolation of the PLSVC. Ideally, this group should have been divided into a subgroup with PLSVC isolation and a subgroup without PLSVC isolation to be able to prove the superiority of PLSVC isolation on maintaining sinus rhythm. I agree that the groups would have been very small.
Author Response
- The authors divided the patients into two groups: PLVSC with triggers (group A) and PLVSC without triggers (group B):
  |
Group A ( n=14) |
Group B ( n=23) |
|
  |
Arrhythmogenic Triggers(+) |
Arrhythmogenic Triggers(-) |
P-value |
Age (years) |
53 ± 8.8 |
72 ± 5.9 |
< 0.001 |
Male, n(%) |
10 (71%) |
15 (62%) |
0.98 |
LA diameter (mm) |
34.3 ± 5.1 |
40.7 ± 4.8 |
0.0011 |
Paroxysmal AF |
12 (86%) |
18 (78%) |
0.89 |
AF duration (months) |
33 ± 22 |
26 ± 20 |
0.35 |
Left ventricular ejection fraction (%) |
62.6 ± 9.1 |
61.5 ± 9.2 |
0.79 |
Diameter of the PLSVC (mm) |
19.1 ± 8.5 |
20.5 ± 6.3 |
0.50 |
Antiarrhythmic drugs, n (%) |
2, (14%) |
4, (17%) |
0.83 |
Amiodarone, n (%) |
0, (0%) |
0, (0%) |
- |
CHADS2 -VASc |
0.77 ± 1.01 |
3.11 ± 1.49 |
< 0.001 |
Redo RFCA |
4 (28%) |
4 (17%) |
0.71 |
TIA/Stroke within 30 days |
0 (0%) |
0 (0%) |
- |
Cardiac tamponade |
0 (0%) |
0 (0%) |
- |
Procedure time (m) |
186 ± 58 |
137 ± 38 |
< 0.001 |
Point 1:.
- The percentage of AF induction in the 2 groups was not described. We only know the percentage of AF initiated by triggers from the PLSVC.
Response 1: Please provide your response for Point 1. (in red)
Thank you for pointing that out. As you pointed out, we didn't show the percentage of AF induction. We retrospectively verified the data again, but no detailed data were available about the induction rate of AF. We would like to use your advice for our next research. We appreciate your advice.
Point 2: The percentage of antiarrhythmic drugs in groups A and B is not known.
Response 2: Please provide your response for Point 2. (in red)
Thank you so much for pointing out the critical issue. According to your suggestion, we have added the percentage of antiarrhythmic drugs in group A and group B. We have changed Table 2 as follows:
Page 6, line177 Table 2
Point 3. Anti-arrhythmic drugs were discontinued for at least five half-lives, with the exception of amiodarone. Patients in group B are older than those in group A and potentially received the most amiodarone. It would be interesting to know the percentage of patients on amiodarone in each group. Group B potentially has more patients on amiodarone which could explain the lack of triggers from the PLSVC.
Response 3: Please provide your response for Point 2. (in red)
Thank you for pointing out. According to your advise, w retrospectively and carefully reviewed the content of oral antiarrhythmic drugs, but none of the patients included in this study took amiodarone. As shown earlier, we added the items about the number of patients taking amiodarone in Table 2. (Page 6, line177, Table 2) Thank you for your advice.
- All patients in group A underwent electrical isolation of the PLSVC. Ideally, this group should have been divided into a subgroup with PLSVC isolation and a subgroup without PLSVC isolation to be able to prove the superiority of PLSVC isolation on maintaining sinus rhythm. I agree that the groups would have been very small.
Response : Please provide your response for Point 2. (in red)
Thank you so much for pointing out a significant issue. I think your point is correct, which is one of this study's limitations. If we were to compare the effects of ablation to triggers more accurately, it would have been more effective to divide all arrhythmogenic PLSVCs into two groups with and without electrical isolation of the PLSVC. The reasons why the study design could not be assembled as you indicated are as follows;
- There were only 14 of 37 PLSVCs with arrhythmogenic triggers in this study.
- All the PLSVC with AF triggers had been ablated because this was a retrospective study.
If we can conduct a prospective study in the future, I would like to design it, as you pointed out.
Thank you so much for your advice.
